# Pemafibrate Ameliorates Steatotic Liver Disease Regardless of Endothelial Dysfunction in Mice

**DOI:** 10.3390/antiox14070891

**Published:** 2025-07-20

**Authors:** Tomoyo Hara, Hiroki Yamagami, Ryoko Uemoto, Akiko Sekine, Yousuke Kaneko, Kohsuke Miyataka, Taiki Hori, Mayuko Ichimura-Shimizu, Masafumi Funamoto, Takeshi Harada, Tomoyuki Yuasa, Shingen Nakamura, Itsuro Endo, Ken-ichi Matsuoka, Yutaka Kawano, Koichi Tsuneyama, Yasumasa Ikeda, Ken-ichi Aihara

**Affiliations:** 1Department of Hematology, Endocrinology and Metabolism, Graduate School of Biomedical Sciences, Tokushima University, 3-18-15 Kuramoto-cho, Tokushima 770-8503, Japan; hara.tomoyo@tokushima-u.ac.jp (T.H.);; 2Department of Community Medicine and Medical Science, Graduate School of Biomedical Sciences, Tokushima University, 3-18-15 Kuramoto-cho, Tokushima 770-8503, Japan; 3Department of Pathology and Laboratory Medicine, Graduate School of Biomedical Sciences, Tokushima University, 3-18-15 Kuramoto-cho, Tokushima 770-8503, Japan; 4Department of Pharmacology, Graduate School of Biomedical Sciences, Tokushima University, 3-18-15 Kuramoto-cho, Tokushima 770-8503, Japan; 5Department of Bioregulatory Sciences, Graduate School of Biomedical Sciences, Tokushima University, 3-18-15 Kuramoto-cho, Tokushima 770-8503, Japan

**Keywords:** MASLD, pemafibrate, eNOS, endothelial dysfunction, oxidative stress, NADPH oxidase

## Abstract

Endothelial dysfunction contributes to the progression of metabolic-dysfunction-associated steatotic liver disease (MASLD). Pemafibrate has been shown to ameliorate MASLD in basic and clinical studies, but it is unclear whether it is also effective in the status of endothelial dysfunction. An MASLD animal model was induced in male wild-type (WT) and endothelial nitric oxide synthase (eNOS)-deficient (eNOSKO) mice by feeding them a high-fat/cholesterol/cholate diet, and they were administered either a vehicle or pemafibrate at 0.17 mg/kg/day for 10 weeks. Although pemafibrate treatment did not change plasma lipid profiles in either WT or eNOSKO mice, pemafibrate reduced plasma AST levels in both WT and eNOSKO mice compared to the levels in the vehicle-treated mice. Histopathological analysis of the liver showed that MASLD was improved in the pemafibrate-treated groups in both WT and eNOSKO mice. Compared to vehicle treatment, pemafibrate treatment significantly reduced the expression levels of hepatic NADPH oxidase subunit genes, M1 macrophages, inflammatory-cytokine-related genes and profibrotic genes in both WT and eNOSKO mice, along with reduction in hepatic oxidative stress assessed by dihydroethidium staining and 4-hydroxynonenal protein levels. Thus, pemafibrate ameliorated MASLD with reduction in oxidative stress and inflammation even in vascular endothelial dysfunction.

## 1. Introduction

Metabolic-dysfunction-associated steatotic liver disease (MASLD) currently affects approximately 40% of the global adult population. The prevalence of MASLD among adults is predicted to rise to over 55% by 2040, which will become the most common cause of chronic liver disease worldwide [1,2,3]. MASLD is recognized as a factor that contributes to the development of liver cirrhosis and liver cancer, as well as major adverse cardiovascular events [4,5]. Therefore, elucidating the detailed pathogenesis of MASLD and the establishment of a method for its prevention are urgently important clinical issues.

Previous experiments using endothelial nitric oxide synthase (eNOS)-deficient mice showed an aggravation of MASLD pathogenesis [6,7], and it was shown that vascular endothelial dysfunction is associated with the severity of MASLD in humans [8]. Therefore, pharmaceutical agents that activate eNOS may ameliorate the pathology of MASLD.

The peroxisome-proliferator-activated receptor α (PPARα) is a PPAR nuclear receptor subfamily member that heterodimerizes with the retinoid X receptor (RXR) and binds to the PPAR response element (PPRE) to activate the transcription of its target genes [9]. PPARα is highly expressed in the liver, brown adipose tissue, heart and kidneys [10]. Most of the target genes of the PPARα are lipid-metabolism-related genes, which are the main targets of exogeneous ligands, including fibrates, for improving hypertriglyceridemia [11]. The PPARα plays a role in MASLD pathogenesis due to its role in β-oxidation in the liver. Hu et al. reported that ubiquity-specific protease 29 stabilizes long-chain alkyl-CoA synthesis 5 expression through K48 deubiquitination and promotes fatty acid activation, which in turn activates PPARα and induces the expression of the genes involved in fatty acid β-oxidation [12].

Pemafibrate is a selective PPARα modulator, and the superiority of pemafibrate compared to classic fibrates has been established by a clinical trial [13]. In addition, despite the fact that the available classic fibrates had an unfavorable impact on the biochemical data of liver and kidney function, pemafibrate showed improvements in liver function and a decreased probability of increasing serum creatinine or decreasing the estimated glomerular filtration rate [13].

Recent clinical and experimental studies have shown that pemafibrate treatment has a favorable effect on improving MASLD, as assessed by non-invasive biomarkers, imaging analysis and animal model analysis [14,15,16,17,18]; however, the molecular mechanisms underlying pemafibrate’s effect on the amelioration of MASLD have not been fully elucidated. Pemafibrate has been shown to exert anti-inflammatory effects in vascular endothelial cells [19] and diminish the levels of vasoconstrictive eicosanoids and free fatty acids, leading to preservation of endothelial function [20]. Furthermore, pemafibrate enhanced ischemia-induced revascularization, which was dependent on eNOS activation [21]. The results together indicate that pemafibrate contributes to inhibiting MASLD progression by activating vascular endothelial function, which is achieved via regulation of the eNOS system. Therefore, the purpose in this study was to clarify the molecular mechanism through which pemafibrate ameliorates the pathology of MASLD and to determine whether this effect depends on vascular endothelial function via the eNOS system.

## 2. Materials and Methods

### 2.1. Animal Preparation

Male 8-week-old wild-type (WT) and eNOS-deficient (eNOSKO) mice of a C57BL/6J genetic background were kept in a specific pathogen-free facility with controlled temperature and a 12 h light/dark schedule, and they were fed with iHFC number 2 diet (68% MF, 28.75% palm oil, 2% cholic acid) with 1.25% cholesterol (HFC1.25) (Hayashi Kasei, Osaka, Japan). The iHFC diet is high in fat, cholesterol and cholate and can induce MASH-related fibrosis, as we reported previously [22,23]. These mice with or without eNOS gene disruption were obtained from The Jackson Laboratory (Sacramento, CA, USA), and they were divided into a vehicle-treated group and a pemafibrate-treated group (n = 8 in each group for a total of 4 groups). Pemafibrate (K-877, Kowa Pharmaceutaical Co., Ltd., Nagoya, Japan) was dissolved in drinking water at a concentration of 1.7 mg/L. In a prior survey of drinking water volume, the average amounts of daily drinking water consumed by both WT and eNOSKO mice were equal at 2.5 mL. The average body weights of WT mice and eNOSKO mice were also the same (25.0 g), which means that the estimated amount of pemafibrate taken per day was approximately 0.17 mg/kg/d. The mice were given either a vehicle or pemafibrate for 10 weeks.

### 2.2. Plasma Biochemical Analysis

Following 10 weeks of vehicle or pemafibrate administration to WT mice and eNOSKO mice (at 18 weeks of age), after overnight fasting, the mice were euthanized, weighed and then subjected to laparotomy to collect blood from the inferior vena cava. The blood was collected into a tube that contained 10% heparin (10 U/mL) and centrifuged at 2000× *g* for 20 min. Plasma was kept in storage at −80 °C until it was required for examination. After blood collection, the liver was immediately excised and weighed. The plasma levels of fasting glucose (FPG) were determined via an enzymatic method using a STAT STRIP XP3 (NIPRO, Osaka, Japan). An enzymatic method was used to analyze the plasma levels of total cholesterol (T-cho), triglycerides (TG), total bilirubin (T-Bil), albumin, aspartate aminotransferase (AST) and alanine aminotransferase (ALT) with a biochemical automated analyzer (BioMajestyTM JCA-BM6050, JEOL Ltd., Tokyo, Japan). In addition, an enzymatic method was used to analyze the plasma levels of high-density lipoprotein cholesterol (HDL-C) with a biochemical automated analyzer (BioMajestyTM JCA-BM8000G, JEOL Ltd., Tokyo, Japan).

### 2.3. Measurements of Hepatic Lipid Contents

For the analysis of hepatic lipids, the liver was freeze-dried, crushed using a TissueLyser II (QIAGEN K.K.—Japan, Tokyo, Japan) with 5 mm stainless-steel beads (QIAGEN K.K.—Japan, catalog number (cat. no.) 69989, Tokyo, Japan), extracted via the Folch method using a chloroform: methanol (2:1) solution, washed with water and dried. The solvent was removed under a nitrogen stream in a fume hood (60 °C), and the tissue was dissolved in isopropanol (FUJIFILM Wako Pure Chemical Corporation, cat. no. 166-04836, Osaka, Japan) containing 10% Nonidet P 40 Substitute (Sigma-Aldrich, cat. no. 74385, Merck KGaA, Darmstadt, Germany). The TG and T-cho concentrations were analyzed via enzymatic methods. Quantitative determination of TG and T-cho was performed in the same manner as that in the plasma biochemistry method described above.

### 2.4. Histopathological Examination of Liver Tissues

For histopathological evaluation, the liver tissues were fixed in 10% phosphate-buffered formalin (FUJIFILM Wako Pure Chemical Corporation, cat. no. 062-01661, Osaka, Japan), and the formalin-fixed liver tissues were embedded in paraffin, sectioned at a thickness of 2 μm and stained with hematoxylin and eosin (HE). In addition, Sirius red staining was used to evaluate liver fibrosis based on the amount of stained pathological collagen. Histopathologic findings of liver steatosis, lobular inflammation and hepatocyte ballooning were evaluated and calculated to determine the NAFLD activity score in accordance with the proposed criteria [24]. Fibrosis staging of the liver tissues was assessed via Sirius red staining.

### 2.5. Immunohistochemistry of Liver Tissues

Immunohistochemistry for the detection of macrophages was performed as previously described [25]. The liver tissues were paraffin-embedded, sectioned at 10 μm in thickness and stained with F4/80 antigen, a rat monoclonal antibody to mouse macrophages (BMA BIOMEDICALS: T-2028. Augst, Switzerland). The distribution of F4/80-positive cells was observed using the avidin–biotin complex technique and Vector Red substrate (Vector Laboratories, Burlingame, CA, USA). Assessment of the extent of macrophage infiltration was determined by calculating the average number of F4/80-positive cells from five separate fields of view for each animal liver tissue.

### 2.6. In Situ Superoxide Detection of Liver Tissues

The dihydroethidium (DHE) method for detecting superoxide production in liver tissue was utilized as previously described [26]. DHE (FUJIFILM Wako Pure Chemical Corporation, cat. no. 041-28251, Osaka, Japan) in PBS (10 μM) was used to incubate non-frozen tissue sections in a dark humidified container at room temperature for 30 min, and they were then examined using a fluorescence microscope (BZ-X800L, KEYENCE, Osaka, Japan).

### 2.7. Gene Expression Analysis of Liver Tissues

Liver total RNA was prepared using TRI Reagent (Cosmo Bio, cat. no. TR118, Tokyo, Japan), and cDNA was synthesized using the QuantiTect^®^ Reverse Transcription Kit (QIAGEN K.K.—Japan, cat. no. 205313, Tokyo, Japan) according to the manufacturer’s instructions. SYBR Green real-time PCR was performed using Power SYBR^TM^ Green PCR Master Mix (Thermo Fisher Scientific Inc., cat. no. 4367659, Tokyo, Japan), specific primers (sequences listed in Appendix A) and a QuantStudio^®^ 3 Real-Time PCR System (Thermo Fisher Scientific Inc., Tokyo, Japan). The amplification process was divided into two stages, one at 50 °C and one at 95 °C, followed by 40 cycles of a 2-step cycle, with 15 s at 95 °C and 30 s at 60 °C. All target gene expression was quantified and adjusted to glyceraldehyde-3-phosphate dehydrogenase (GAPDH).

### 2.8. Western Blot Analysis of Liver Tissues

Total protein was extracted from the homogenized liver tissues using the Thermo Scientific T-PER Tissue Protein Extraction Reagent (Thermo Fisher Scientific Inc., cat. no. 78510, Tokyo, Japan) with protease inhibitors (PhosSTOP^™^, cat. no. 04906845001, and cOmplete^TM^, cat. no.11836170001, Merck KGaA, Darmstadt, Germany). In brief, 10 μg of liver protein that was extracted was used for Western blotting after boiling for 5 min in Laemmli sample buffer (FUJIFILM Wako Pure Chemical Corporation, cat. no. 198-13282, Osaka, Japan). The liver protein extracts were subjected to SDS-PAGE and then transferred to a PVDF membrane (Merck KGaA, cat. no. IPVH00010, Darmstadt, Germany). The membrane underwent 20 min of blockage at room temperature using SuperBlock T20 TBS Blocking Buffer (Thermo Fisher Scientific K.K., cat. no. 21059, Tokyo, Japan). Each primary antibody was added to the blots overnight at 4 °C, followed by 1 h incubation with an anti-rabbit secondary antibody (conjugated with horseradish peroxidase) (Cell Signaling Technology (CST), Inc. cat. no. 7074, Danvers, MA, USA). Enhanced chemiluminescence with Amersham ECL Prime Western Blotting Detection Reagent (Cytiva, cat. no. RPN2236, Tokyo, Japan) and exposure to the ChemiDoc Touch MP imaging system 17001402JA (Bio-Rad Laboratories, Inc. 1000 Alfred Nobel Drive, Hercules, CA, USA) were used to visualize immunoreactive bands. We used primary antibodies against Akt (Cell Signaling Technology (CST), Inc. cat. no. 9272, Danvers, MA, USA), phosphorylated Akt (Ser473) (CST, Inc., cat. no. 9271, Danvers, MA, USA), AMPKα (CST, Inc., cat. no. 2603, Danvers, MA, USA), phosphorylated AMPKα (Thr172) (CST, Inc., cat. no. 2535, Danvers, MA, USA), glyceraldehyde-3-phosphate dehydrogenase (GAPDH) (CST, Inc., cat. no. 2118, Danvers, MA, USA) and 4-hydroxynonenal (4-HNE) (JaICA/NIKKEN SEIL. Co, Ltd., cat. no.MHN-100P, Shizuoka, Japan). Densitometry was performed for signal quantification using ImageJ version 1.54g (https://imagej.net/ij/index.html accessed on 8 January 2025). Eight independent samples in each group were analyzed, and GAPDH was used as an internal control. The molecular-weight marker used was the ECL^TM^ Rainbow^TM^ Marker-Full Range (Cytiva, cat. no. RPN800E, Tokyo, Japan).

### 2.9. Statistical Analysis

Dot plots with mean bars depict the values for every parameter in the group. Quantitative data were compared among groups, and the Kruskal–Wallis test with Dunn’s multiple comparisons test were used to assess statistical significance. These analyses were performed using GraphPad Prism 10 (GraphPad Software, Boston, MA, USA). The threshold for statistical significance was set to *p* < 0.05.

## 3. Results

### 3.1. Body Weight, Liver Weight and Biochemical Analysis

As shown in Figure 1a,b, there were no significant differences among any of the groups in terms of body weight and liver weight.

In the liver function tests, there was no significant difference in T-Bil levels among the four groups of mice (Figure 1c); pemafibrate increased the plasma albumin levels in eNOSKO mice (Figure 1d); and AST was reduced by pemafibrate treatment regardless of eNOS gene deficiency (Figure 1e), but ALT was significantly reduced by pemafibrate treatment only in WT mice (Figure 1f). There was a significant difference in FPG levels between pemafibrate-treated WT mice and vehicle-treated eNOSKO mice (Figure 1g). No significant difference in the plasma lipid profile, including T-cho, TG and HDL-C, was found among the mouse groups (Figure 1h–j). Regarding the hepatic contents of TG and T-cho, significant differences were observed only between the pemafibrate-treated WT mice and the vehicle-treated eNOSKO mice (Figure 1k,l).

### 3.2. Pemafibrate Ameliorates Steatotic Lesions and Attenuates Macrophage Infiltration in the Liver Regardless of eNOS

Representative histopathological findings of the liver are shown in Figure 2a. The assessment of each hepatic pathological finding, including steatosis, lobular inflammation and ballooning of hepatocytes, only showed significant difference in steatosis between WT mice with and without pemafibrate treatment. However, the NAFLD activity score, as a comprehensive index of MASLD, showed that pemafibrate treatment significantly reduced the scores in both WT and eNOSKO mouse groups compared to those in both vehicle treatment groups. The findings based on Sirius red staining demonstrated that pemafibrate treatment significantly reduced the areas of liver fibrosis in both WT and eNOSKO mouse groups compared to those in both vehicle treatment groups (Figure 2a,c).

Immunohistochemical analyses showed that F4/80-positive cell counts were significantly and similarly decreased in the pemafibrate-treated groups compared to those in the vehicle-treated groups in both WT mice and eNOSKO mice (Figure 2a,d).

### 3.3. Pemafibrate Attenuates Hepatic Production of Reactive Oxygen Species Regardless of eNOS

We evaluated the production of reactive oxygen species (ROS) by the DHE fluorescent intensities in the mouse groups. As shown in Figure 2a,e, the DHE fluorescent intensities were significantly reduced by pemafibrate treatment compared to vehicle treatment in both WT mice and eNOSKO mice. The results suggest that pemafibrate can inhibit the production of ROS in MASLD regardless of eNOS deficiency.

### 3.4. Quantitative PCR Analysis of MASLD-Associated Genes in the Liver

The candidate genes of MASLD-associated factors and each primer sequence for quantitative PCR analysis are listed in Appendix A.

#### 3.4.1. Pemafibrate Attenuates Expression of NADPH Oxidase Subunit Genes Regardless of eNOS

Since activation of the nicotinamide adenine dinucleotide phosphate (NADPH) oxidase system is a crucial source of superoxide production, we assessed the gene expression levels of NADPH oxidase subunits. As shown in Figure 3 and Appendix A, no significant alterations in expression levels were found in *Rac1*, *Ncf1*, *Noxo1*, *Nox1* and *Nox4* genes. However, the gene expression levels of *Ncf2*, *Ncf4*, *Cyba* and *Cybb* were significantly reduced in the pemafibrate-treated groups compared to those in the vehicle-treated groups in both WT mice and eNOSKO mice.

#### 3.4.2. Pemafibrate Attenuates Expression of M1 Macrophages and Macrophage-Associated Genes Regardless of eNOS

It is well known that M1 macrophages have proinflammatory activity and that M2 macrophages are responsible for the resolution of inflammation. The expression levels of M1 macrophage indicator genes, including *Ccl2*, *Adgre1* and *Itgax*, were significantly lower in the pemafibrate-treated groups than in the vehicle-treated groups regardless of eNOS deficiency (Figure 3). In addition, the expression levels of M1 macrophage-associated genes, including *Cxcl1*, *Cxcl2* and *Clec4e*, were significantly lower in the pemafibrate-treated groups than those in the vehicle-treated groups regardless of eNOS deficiency (Figure 3). On the other hand, the expression levels of M2 macrophage-associated genes, including *Cd163*, *Mrc1* and *Cd209a*, were not different among the groups (Appendix A).

#### 3.4.3. Pemafibrate Attenuates Expression of Inflammatory Cytokine, Chemokine and Profibrotic Genes Regardless of eNOS

Since the development of MASLD is linked to the recruitment of proinflammatory cells, such as M1 macrophages and other myeloid lineage cells, and the initiation of inflammation via secretion of effector-type cytokines and chemokines [27], we evaluated the expression of inflammatory cytokine- and chemokine-associated genes. As shown in Figure 3, the gene expression levels of *Il1b*, *Tnf* and *Ifnγ* were reduced in the pemafibrate-treated groups compared to those in the vehicle-treated groups in both WT mice and eNOSKO mice.

Since tissue fibrosis progresses with increased inflammation, we evaluated the expression levels of genes encoding tissue fibrosis promoters. The expression levels of *Tgfb1*, *Col1a1* and *Col3a1* were significantly lower in the pemafibrate-treated groups than those in the vehicle-treated groups in both WT mice and eNOSKO mice (Figure 3).

#### 3.4.4. Other Candidate Genes

As shown in Appendix A, genes involved in the redox system, including *Nfe2l2*, *Keap- 1* and *Hmox1*, did not appear to be associated with the pathological modification of MASLD in this study. There was no significant difference in the gene expression levels of *PPARα* among the mouse groups.

#### 3.4.5. Analysis of Gene Expression Clustergrams and Volcano Plots Shows Comprehensive Anti-MASLD Effects of Pemafibrate Regardless of eNOS

A whole-sample gene expression clustergram (Appendix A) and a group-wise integrated gene expression clustergram (Figure 4a) showed the summary of pemafibrate-induced alterations of gene expression profiles in WT mice and eNOSKO mice. The clustergrams revealed that pemafibrate suppressed the gene expression of NADPH oxidase subunits, M1 macrophages and their related factors, inflammatory cytokines and chemokines and profibrotic factors regardless of eNOS deficiency. As shown in Figure 4b, analyses of the volcano plots with reference to vehicle-treated WT mice showed that pemafibrate suppressed the MASLD-promoting genes in both WT mice and eNOSKO mice in a similar way.

### 3.5. Pemafibrate Activates Hepatic AMPKα and Attenuates 4-HNE Regardless of eNOS

Since activation of the Akt and AMP-activated protein kinase (AMPK) signaling pathway has been shown to be associated with the development of MASLD [28], we examined those factors via Western blot analysis. As shown in Figure 5a, although Akt phosphorylation was not different among the mouse groups, AMPKα phosphorylation was significantly greater in the pemafibrate-treated groups than in the vehicle-treated groups regardless of eNOS deficiency (Figure 5b). Since lipid peroxidation produces 4-HNE, which has become widely accepted as a cause of oxidative stress, we estimated hepatic 4-HNE and found that 4-HNE levels were significantly lower in the pemafibrate-treated groups than those in the vehicle-treated groups regardless of eNOS deficiency (Figure 5c). These results are consistent with the results for DHE fluorescence intensities in this study.

## 4. Discussion

In this study, we found that pemafibrate reduced oxidative stress and the expression levels of inflammatory cytokines and chemokines and that pemafibrate ameliorated iHFC diet-induced MASLD in mice with and without eNOS deficiency.

### 4.1. Pemafibrate Attenuates Hepatic Oxidative Stress Regardless of eNOS

The cellular state of oxidative stress involves the production of harmful molecules, such as ROS, which can lead to reduced cell viability and promote cell death [29]. ROS are capable of promoting necrotic inflammation and activating various intracellular signaling pathways, leading to hepatocyte apoptosis [2]. Since an increase in hepatic cell death via apoptosis has been observed in individuals and experimental animal models of MASLD, the development and progression of MASLD are thought to be associated with a pathological increase in cell death in the liver and peripheral tissues [30].

The role of pemafibrate in reducing oxidative stress in various metabolic and cardiovascular disorders has not been fully elucidated, despite previous studies showing the ability of pemafibrate to reduce oxidative stress in the cardiovascular and renal systems [31,32,33]. Maki et al. reported that in obese diabetic *db*/*db* mice, increased expression of *Nox4* in the kidneys was accompanied by increases in oxidative stress markers, such as urinary 8OHdG and DHE fluorescence, and that pemafibrate reduced these oxidative stress markers along with suppression of *Nox4* expression in the kidney [31]. Similarly, in our study, the administration of pemafibrate suppressed oxidative stress as assessed by DHE fluorescence intensities and 4-HNE protein levels in the liver tissue of mice with and without eNOS gene disruption. However, as components of the NADPH oxidase system, there was no alteration in the gene expression of hepatic *Nox4* in MASLD mice in this study (Appendix A). Instead, pemafibrate suppressed the expression levels of *Ncf2*, *Ncf4*, *Cyba* and *Cybb* as components of the NADPH oxidase system in our MASLD mice regardless of eNOS deficiency (Figure 3). The results showed that the NADPH oxidase system unit genes may be differently activated and inhibited due to differences in metabolic and/or physical stimuli, organ specificity, the presence or absence of PPRE in each subunit promoter region, the recruitment of coactivators involved in activating PPARα and the effective pemafibrate concentration. Further detailed studies are required to verify this hypothesis. In this study, pemafibrate improved MASLD despite almost no significant changes in the lipid profile. However, since the inhibitory effect of the NADPH oxidase system through activation of PPARα has been observed in studies other than studies using metabolic disorder models [34,35], pemafibrate may have a direct inhibitory effect on the production of ROS via inactivation of the NADPH oxidase system, independent of metabolic disorders, including lipid metabolism.

### 4.2. Pemafibrate Attenuates Hepatic Inflammatory Cytokines and Chemokines Regardless of eNOS

The activation of hepatic macrophages (Kupffer cells) has been proposed to play a key role in driving the development of MASLD, since activated macrophages are thought to activate hepatic stellate cells (HSCs), leading to their trans-differentiation into collagen-producing myofibroblasts, which drive liver fibrosis [36,37]. Upregulation of proinflammatory cytokines, including *Tnf*, *Il1b* and *Il6*, and chemokines, such as *Ccl2*, is often a result of this activation, which increases monocyte and neutrophil infiltration and fosters an inflammatory milieu in the MASLD liver [37,38]. The activation of macrophages has been suggested to boost MASLD progression by either exaggerating liver steatosis via inhibition of the PPARα signaling pathway in hepatocytes or by promoting the activation of HSCs through enhancement of *Tnf* and *Il1b* [39,40]. These reports are consistent with the results of our study, showing that pemafibrate inhibited the expression of inflammatory cytokine- and chemokine-related genes and ameliorated MASLD.

### 4.3. Pemafibrate Activates Hepatic AMPK Regardless of eNOS

AMPK, as a crucial kinase widely expressed in various hepatic cells, has been shown to have multiple organ protective effects. Since AMPK regulates mitochondrial function and cell death pathways to counteract hepatocyte injury and maintain hepatocyte energy balance, the activation of AMPK suppresses the production of proinflammatory mediators and cytokines, leading to inhibition of the development of liver fibrosis [41,42,43,44]. In addition, recent research has shown that hepatic AMPK deficiency exacerbates hepatocyte death and liver injury in an animal model of liver fibrosis caused by a cholecalciferol-deficient high-fat diet [41]. Thus, it is probable that AMPK plays an important role in the modification of MASLD. Pemafibrate activates the phosphorylation of AMPK in the renal tissue of diabetic *db*/*db* mice [31]. The glucocorticoid receptor and PPARα cooperatively activate a lipid catabolic gene program in primary hepatocytes through direct recruitment of the activated AMPK [45]. The results of the analyses of the protein–protein interaction network, molecular docking, functional enrichment and Western blotting showed that isoxanthohumol, the major hop flavonoid, activated the hepatic phosphorylation of AMPK and PPARα, with the PI3/Akt signaling pathways activated in a hyperlipidemic mouse model [46]. These results suggest that the amelioration of MASLD observed in our study was partly due to reciprocal reactions between phosphorylated AMPKα and activated PPARα via pemafibrate stimulation.

### 4.4. Unresolved Issues

In our previous studies of cardiovascular and renal dysfunction in animal models using eNOSKO mice, the mice showed a clearly worsening phenotype of organ damage compared to that in WT mice [47,48,49]. In addition, in MASLD animal models, it has been reported that vascular endothelial dysfunction models, such as a model with eNOS deficiency, show progression of liver damage [6,7]. Based on these findings, we predicted that eNOS deficiency would cause the progression of MASLD in our animal model [6,7,8]. However, in this study, the severity of MASLD in eNOSKO mice was surprisingly similar to that in WT mice. There are various methods for creating MASLD in animal models, including the use of various drugs, special diets, and it has been reported that the phenotypes differ depending on the animal species, strain, diet and pharmaceutical stimuli [50,51]. The special diet (iHFC) we used in this study was established for the purpose of creating a pathological model of MASLD [22,23]. We previously observed that iHFC feeding caused MASLD along with mitochondrial dysfunction in rats [52], and mitochondrial dysfunction has also been reported in the development of MASLD in eNOSKO mice [6]. Therefore, it is possible that the progression mechanism of the iHFC diet-induced MASLD in both WT mice and eNOSKO mice is partly common in mitochondrial dysfunction. This may also be the reason why no significant difference was observed in the severity of MASLD induced by iHFC feeding between WT mice and eNOSKO mice.

Although we reviewed previous studies, we found no strong evidence that a 10-week pemafibrate intervention from 8 to 18 weeks of age was appropriate for our study design. Moreover, whether earlier pemafibrate intervention is more effective in ameliorating MASLD regardless of eNOS gene disruption should be verified. In addition, since the promotion of ischemia-induced revascularization by pemafibrate was dependent on eNOS activation [21], the eNOS-independent liver protective effect of pemafibrate observed in our study is an interesting result that could lead to elucidation of a novel molecular mechanism of pemafibrate for promoting health science. Thus, further research is required to clarify these unsolved issues in detail.

### 4.5. Experimental Limitations

The following concerns regarding the study methods and analysis should be considered: First, the drinking water was individually monitored, as shown in Appendix A. We do not think that the drug exposure considerably differed among individuals. However, the forced oral administration of a fixed drug dose during the experiment would have ensured consistent drug exposure. Therefore, oral gavage with a fixed pemafibrate dose should be used in a future study to further increase the reliability and reproducibility of our results and minimize interindividual data variability.

Second, the results of a power analysis to ensure the detection of differences in the data among the four groups showed that a sample size of 45 mice per group was required (Appendix A); therefore, the statistical power of our study is limited due to the small sample size used.

## 5. Conclusions

In conclusion, pemafibrate, a selective PPARα modulator, ameliorates diet-induced MASLD with reduction in oxidative stress regardless of eNOS gene disruption in mice. This result suggests that pemafibrate might be a useful agent for the treatment of MASLD patients with dyslipidemia and endothelial dysfunction, which is a crucial cardiovascular risk. Further large-scale clinical studies are also warranted to conclude this clinical question.

## Figures and Tables

**Figure 1 antioxidants-14-00891-f001:**
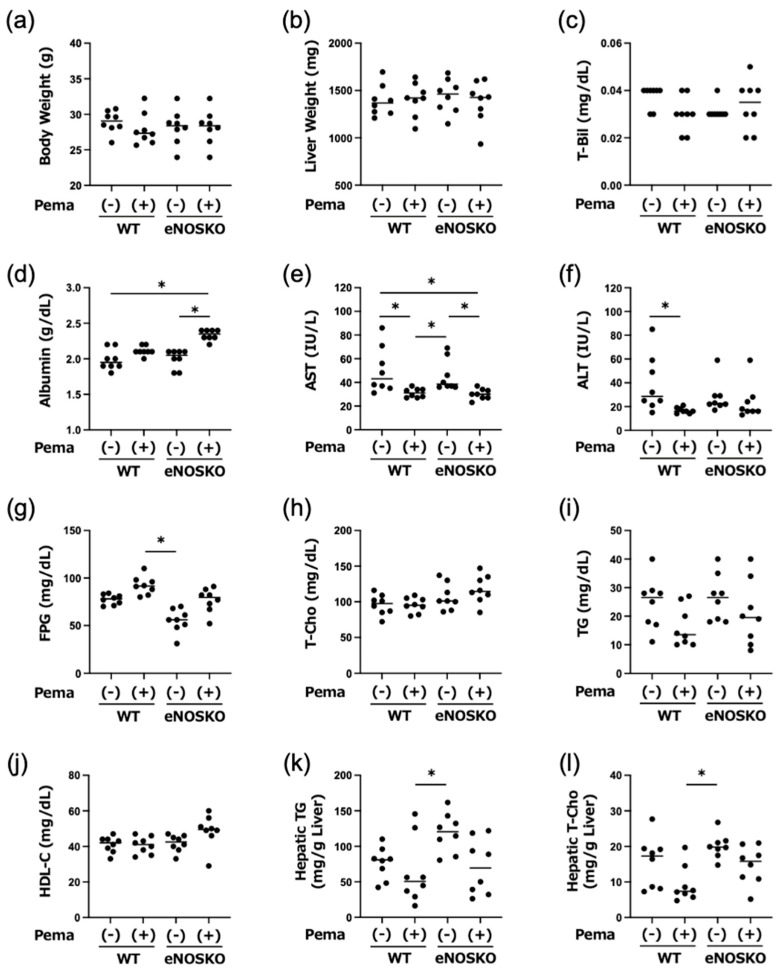
Body weight (**a**), liver weight (**b**) and results of biochemical analyses (**c**–**l**) of plasma samples and liver tissues in vehicle-treated MASLD mice and pemafibrate-treated MASLD mice with and without eNOS deficiency. Bars represent the mean values in each group (n = 8 mice per experimental group). * *p* < 0.05 is presented between the indicated groups. Kruskal–Wallis test with Dunn’s multiple comparisons test were used to assess statistical significance.

**Figure 2 antioxidants-14-00891-f002:**
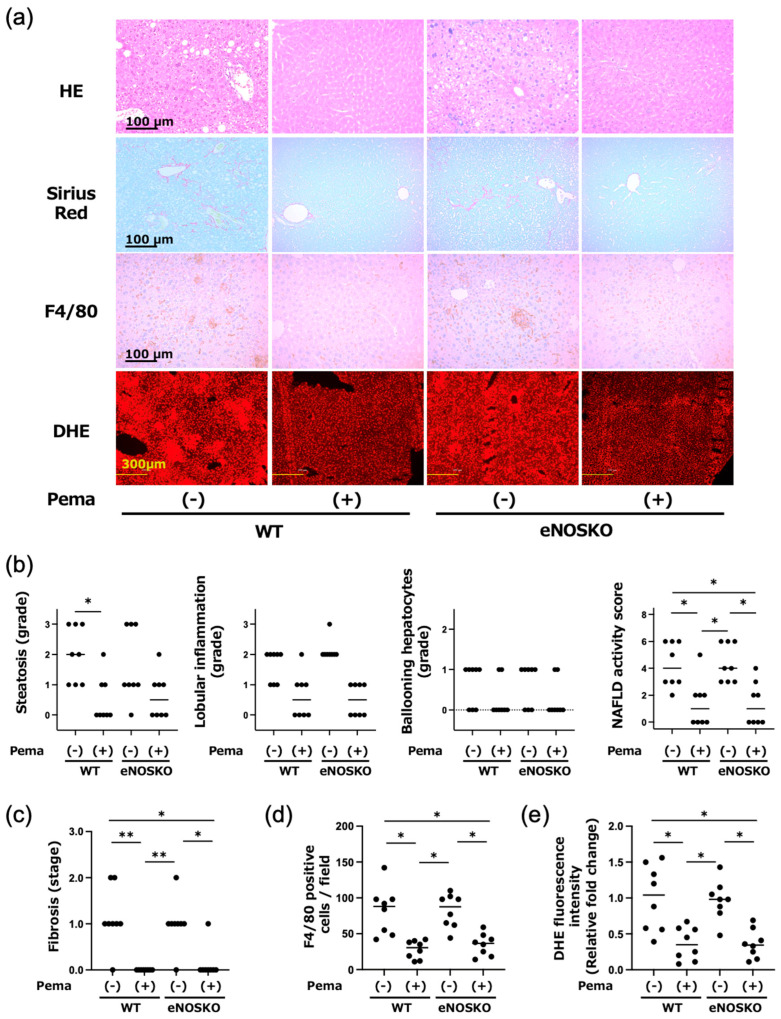
Hepatic histopathological examination, macrophage infiltration and superoxide detection in vehicle-treated MASLD mice and pemafibrate-treated MASLD mice with and without eNOS deficiency. (**a**) Representative findings of HE, Sirius red and F4/80 antibody staining and fluorescence with DHE in the liver of vehicle-treated MASLD mice and pemafibrate-treated MASLD mice with and without eNOS deficiency. (**b**) Histopathological assessment of the severity of MASLD in liver tissues. (**c**) Comparison of fibrosis staging in liver tissues. (**d**) Quantification of average F4/80-positive cell numbers counted in five independent visual fields in liver tissues. (**e**) Semi-quantitative analysis of DHE fluorescence intensities of liver tissues. Bars represent the mean values in each group (n = 8 mice per experimental group). * *p* < 0.05 and ** *p* < 0.01 are presented between the indicated groups. Kruskal–Wallis test with Dunn’s multiple comparisons test were used to assess statistical significance.

**Figure 3 antioxidants-14-00891-f003:**
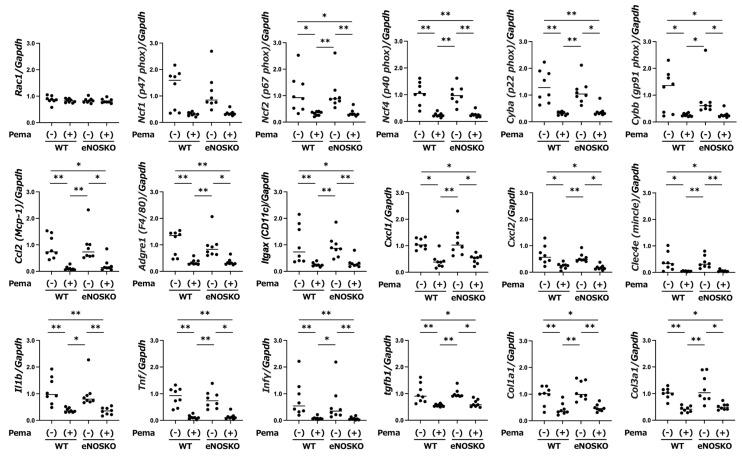
Hepatic gene expression levels of MASLD-associated factors in vehicle-treated MASLD mice and pemafibrate-treated MASLD mice with and without eNOS deficiency. Bars represent the mean values in each group (n = 8 mice per experimental group). * *p* < 0.05 and ** *p* < 0.01 are presented between the indicated groups. Kruskal–Wallis test with Dunn’s multiple comparisons test were used to assess statistical significance.

**Figure 4 antioxidants-14-00891-f004:**
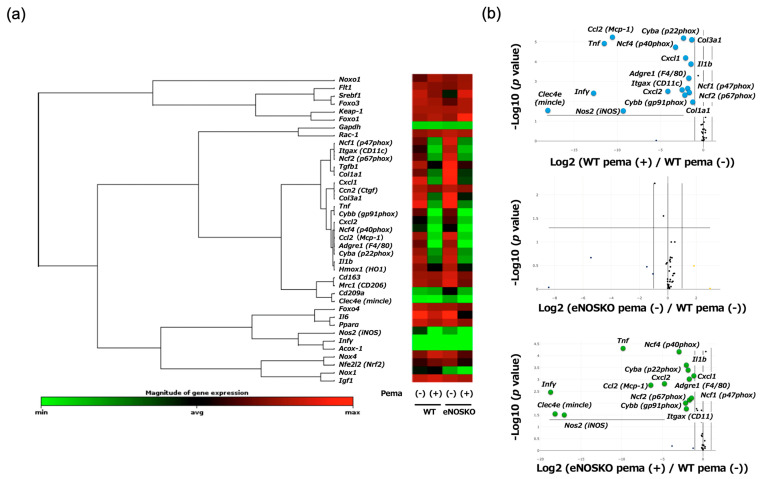
Profiling of hepatic gene expression in vehicle-treated MASLD mice and pemafibrate-treated MASLD mice with and without eNOS deficiency. (**a**) Group-wise integrated gene expression clustergram. (**b**) Volcano plots with reference to the vehicle-treated WT mouse group. n = 8 mice per experimental group.

**Figure 5 antioxidants-14-00891-f005:**
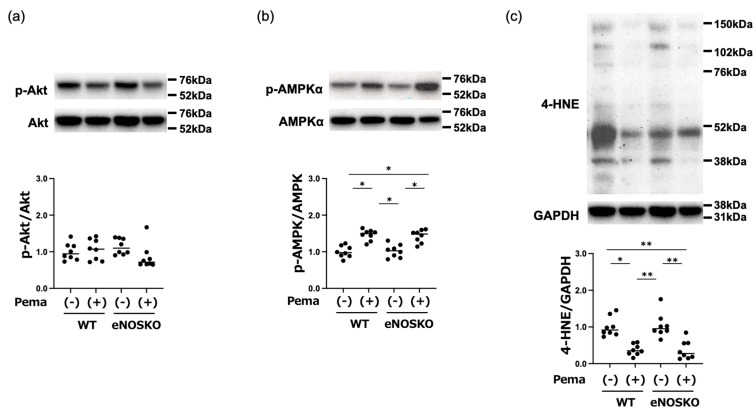
Western blot analysis of MASLD-modulating factors in liver tissues of vehicle-treated MASLD mice and pemafibrate-treated MASLD mice with and without eNOS deficiency. (**a**) Representative blots of phosphorylated Akt and Akt, and ratios of phosphorylated Akt to Akt. (**b**) Representative blots of phosphorylated AMPKα and AMPKα, and ratios of phosphorylated AMPKα to AMPKα. (**c**) Representative blots of 4-HNE and GAPDH, and ratios of 4-HNE to GAPDH. Bars represent the mean values in each group (n = 8 mice per experimental group). * *p* < 0.05 and ** *p* < 0.01 are presented between the indicated groups. Kruskal–Wallis test with Dunn’s multiple comparisons test were used to assess statistical significance.

## Data Availability

We have no additional data available to share.

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
