# Peer review of "Pemafibrate Ameliorates Steatotic Liver Disease Regardless of Endothelial Dysfunction in Mice"

_antioxidants, 2025, doi:10.3390/antiox14070891_

Round 1

Reviewer 1 Report

This investigation contains some issues that need to be carefully addressed before further steps.

  1. The tittle of the investigation is too long, “Pemafibrate, a Selective Peroxisome Proliferator-Activated Receptor α Modulator, Ameliorates Steatotic Liver Disease with Reduction of Oxidative Stress Regardless of Endothelial Dysfunction in Mice”, the description of pemafibrate role is irrelevant in the title. That information can be described in detail in either the introduction, materials and methods or discussion section.
  2. 4-HNE is not a protein; is a highly reactive aldehyde generated as a product of lipid peroxidation that binds to proteins as an adduct; however, authors show this aldehyde as a WB protein band (figure 5b). When determining the presence of 4-HNE adducts, the entire lanes of the WB analysis should be included, not just a single band, since 4-HNE binds to many proteins, and the quantification of the adducts should be performed from the entire WB lanes.
  3. Subtitles of result sections read as methods subtitles. Subtitles of result section should summarize the main finding/conclusion of the subsection and they shall be attractive for readers and be written in present tense. This because they are showing the main findings of the investigation, which means that if someone else performs the same experiments by using the same models with the same variables, they will get the same results. Actually, they should be written as partial conclusions of the investigation. For example, authors wrote “3.5. Western Blot Analysis”, instead, the subtitle should state the relevant finding of that subsection.
  4. Kits and reagents should contain the full vendor information, such as catalogue number, company, place and country of production. Authors should include this information throughout materials and methods section as some kits/reagents do not show that information.
  5. For an easier validation of the evaluated gene by readers, primers shown in Table S1 should include their respective NCBI Reference Sequence.

N/A

Author Response

Reviewer 1

We thank Reviewer 1 for your careful review of our research and precise advice. Following reviewer 1's comments, we have provided our responses.

This investigation contains some issues that need to be carefully addressed before further steps.

  1. The tittle of the investigation is too long, “Pemafibrate, a Selective Peroxisome Proliferator-Activated Receptor α Modulator, Ameliorates Steatotic Liver Disease with Reduction of Oxidative Stress Regardless of Endothelial Dysfunction in Mice”, the description of pemafibrate role is irrelevant in the title. That information can be described in detail in either the introduction, materials and methods or discussion section.

(Response)  Thank you for this feedback. We have simplified the title to “Pemafibrate Ameliorates Steatotic Liver Disease Regardless of Endothelial Dysfunction in Mice”.

  1. 4-HNE is not a protein; is a highly reactive aldehyde generated as a product of lipid peroxidation that binds to proteins as an adduct; however, authors show this aldehyde as a WB protein band (figure 5b). When determining the presence of 4-HNE adducts, the entire lanes of the WB analysis should be included, not just a single band, since 4-HNE binds to many proteins, and the quantification of the adducts should be performed from the entire WB lanes.

(Response)  In accordance with this suggestion, we have reperformed the Western blot experiment and requantified the band densities by considering the entire lane, and Figure 5 has been revised to provide these results.

  1. Subtitles of result sections read as methods subtitles. Subtitles of result section should summarize the main finding/conclusion of the subsection and they shall be attractive for readers and be written in present tense. This because they are showing the main findings of the investigation, which means that if someone else performs the same experiments by using the same models with the same variables, they will get the same results. Actually, they should be written as partial conclusions of the investigation. For example, authors wrote “3.5. Western Blot Analysis”, instead, the subtitle should state the relevant finding of that subsection.

(Response)   Thank you for this valuable advaice. We have modified the subsection headings to provide a concise description of the results in that subsection.

  1. Kits and reagents should contain the full vendor information, such as catalogue number, company, place and country of production. Authors should include this information throughout materials and methods section as some kits/reagents do not show that information.

(Response)   To address the reviewer’s comment, we have now provided the information for all commercial products used in our study.

  1. For an easier validation of the evaluated gene by readers, primers shown in Table S1 should include their respective NCBI Reference Sequence.

(Response)   Thank you for this feedback. We have included the NCBI Reference Sequence ID for every primer in the revised Table S1.

Reviewer 2 Report

Topic is very interesting. : Endothelial dysfunction contributes to the progression of metabolic dysfunction- 
associated steatotic liver disease (MASLD). In addition, Pemafibrate has been shown to ameliorate MASLD in 
basic and clinical studies, but it is unclear whether it is also effective in the status of endothelial 
dysfunction. 

Study design includes an MASLD animal model was induced in male wild-type (WT) and endothelial nitric oxide synthase (eNOS)-deficient (eNOSKO) mice by feeding them a high-fat/cholesterol/cholate diet, 
and they were administered either a vehicle or pemafibrate at 0.17 mg/kg/day for 10 weeks. 
Although pemafibrate treatment did not change plasma lipid profiles in either WT or eNOSKO mice, 
pemafibrate reduced plasma AST levels in both WT and eNOSKO mice compared to the levels in 
the vehicle-treated mice. Histopathological analysis of the liver showed that MASLD was improved 
in the pemafibrate-treated groups in both WT and eNOSKO mice. Compared to vehicle treatment,
pemafibrate treatment significantly reduced the expression levels of hepatic NADPH oxidase 
subunit genes, M1 macrophages, inflammatory cytokine-related genes and profibrotic genes in both 
WT and eNOSKO mice along with reduction of hepatic oxidative stress assessed by dihydroethidium staining and 4-hydroxynonenal protein levels. Thus, pemafibrate ameliorated MASLD with reduction of oxidative stress and inflammation even in vascular endothelial dysfunction. All figures are comprehensive. All figures are comprehensive. References are well chosen.

Author Response

Reviewer 2

We thank you for your time in reviewing our manuscript and for your positive evaluation.

Topic is very interesting. : Endothelial dysfunction contributes to the progression of metabolic dysfunction- associated steatotic liver disease (MASLD). In addition, Pemafibrate has been shown to ameliorate MASLD in basic and clinical studies, but it is unclear whether it is also effective in the status of endothelial dysfunction. 

Study design includes an MASLD animal model was induced in male wild-type (WT) and endothelial nitric oxide synthase (eNOS)-deficient (eNOSKO) mice by feeding them a high-fat/cholesterol/cholate diet, and they were administered either a vehicle or pemafibrate at 0.17 mg/kg/day for 10 weeks.

Although pemafibrate treatment did not change plasma lipid profiles in either WT or eNOSKO mice, pemafibrate reduced plasma AST levels in both WT and eNOSKO mice compared to the levels in the vehicle-treated mice. Histopathological analysis of the liver showed that MASLD was improved in the pemafibrate-treated groups in both WT and eNOSKO mice. Compared to vehicle treatment, pemafibrate treatment significantly reduced the expression levels of hepatic NADPH oxidase subunit genes, M1 macrophages, inflammatory cytokine-related genes and profibrotic genes in both WT and eNOSKO mice along with reduction of hepatic oxidative stress assessed by dihydroethidium staining and 4-hydroxynonenal protein levels. Thus, pemafibrate ameliorated MASLD with reduction of oxidative stress and inflammation even in vascular endothelial dysfunction. All figures are comprehensive. References are well chosen.

Reviewer 3 Report

The manuscript presents an interesting investigation into pemafibrate's effects on MASLD in both wild-type and eNOS-deficient mice. However, several fundamental issues need to be addressed:

  1. In the introduction section, elaborate on the MASLD disease burden by addressing the most updated epidemiology of MASLD (PMID: 39159948). 
  2. The authors acknowledge that eNOSKO mice did not show the expected worsened MASLD phenotype compared to WT mice. This finding contradicts their initial hypothesis and undermines the study's primary rationale. While they provide potential explanations (common mitochondrial dysfunction mechanisms), this major deviation from expected results requires more thorough investigation and discussion of its implications for the study's conclusions.
  3. The study uses n=8 per group, but no power calculation or sample size justification is provided. Given the unexpected lack of difference between WT and eNOSKO mice, a post-hoc power analysis would be valuable to determine if the study was adequately powered to detect meaningful differences.
  4. The pemafibrate dosing (0.17 mg/kg/day) is based on estimated water consumption and body weight averages. This approach introduces variability in actual drug exposure between animals. Individual monitoring of water consumption or alternative delivery methods (e.g., gavage) would provide more precise dosing.
  5. The 10-week treatment period begins at 8 weeks of age, ending at 18 weeks. Justification for this specific timing and duration is lacking. Earlier intervention might have revealed different outcomes, particularly given that MASLD pathogenesis begins early in this model.
  6. The 10-week treatment period begins at 8 weeks of age, ending at 18 weeks. Justification for this specific timing and duration is lacking. Earlier intervention might have revealed different outcomes, particularly given that MASLD pathogenesis begins early in this model.
  7. The selective reduction in Ncf2, Ncf4, Cyba, and Cybb expression without changes in Nox1, Nox4, Rac1, Ncf1, and Noxo1 is intriguing but requires a deeper mechanistic explanation. The authors should discuss why only specific NADPH oxidase subunits were affected and the functional implications of this selective regulation.
  8. While pemafibrate increased AMPK phosphorylation regardless of eNOS status, the upstream mechanisms leading to this activation remain unclear. The relationship between PPARα activation and AMPK phosphorylation in this context needs better elucidation.
  9. The use of the Kruskal-Wallis test with Dunn's multiple comparisons is appropriate for non-parametric data, but the manuscript lacks information about data distribution testing and assumptions validation.
  10. To speculate on the mechanisms underlying the protective role of PPARα agonists in MASLD, address the following recent reference (PMID: 39355870, page 163: " USP29 stabilized ACSL5 expression to promote the activation of fatty acids, which in turn activated PPARa to induce the expression of genes involved in FAO and then directed them to the mitochondria for β-oxidation.").

The manuscript presents an interesting investigation into pemafibrate's effects on MASLD in both wild-type and eNOS-deficient mice. However, several fundamental issues need to be addressed:

  1. In the introduction section, elaborate on the MASLD disease burden by addressing the most updated epidemiology of MASLD (PMID: 39159948). 
  2. The authors acknowledge that eNOSKO mice did not show the expected worsened MASLD phenotype compared to WT mice. This finding contradicts their initial hypothesis and undermines the study's primary rationale. While they provide potential explanations (common mitochondrial dysfunction mechanisms), this major deviation from expected results requires more thorough investigation and discussion of its implications for the study's conclusions.
  3. The study uses n=8 per group, but no power calculation or sample size justification is provided. Given the unexpected lack of difference between WT and eNOSKO mice, a post-hoc power analysis would be valuable to determine if the study was adequately powered to detect meaningful differences.
  4. The pemafibrate dosing (0.17 mg/kg/day) is based on estimated water consumption and body weight averages. This approach introduces variability in actual drug exposure between animals. Individual monitoring of water consumption or alternative delivery methods (e.g., gavage) would provide more precise dosing.
  5. The 10-week treatment period begins at 8 weeks of age, ending at 18 weeks. Justification for this specific timing and duration is lacking. Earlier intervention might have revealed different outcomes, particularly given that MASLD pathogenesis begins early in this model.
  6. The 10-week treatment period begins at 8 weeks of age, ending at 18 weeks. Justification for this specific timing and duration is lacking. Earlier intervention might have revealed different outcomes, particularly given that MASLD pathogenesis begins early in this model.
  7. The selective reduction in Ncf2, Ncf4, Cyba, and Cybb expression without changes in Nox1, Nox4, Rac1, Ncf1, and Noxo1 is intriguing but requires a deeper mechanistic explanation. The authors should discuss why only specific NADPH oxidase subunits were affected and the functional implications of this selective regulation.
  8. While pemafibrate increased AMPK phosphorylation regardless of eNOS status, the upstream mechanisms leading to this activation remain unclear. The relationship between PPARα activation and AMPK phosphorylation in this context needs better elucidation.
  9. The use of the Kruskal-Wallis test with Dunn's multiple comparisons is appropriate for non-parametric data, but the manuscript lacks information about data distribution testing and assumptions validation.
  10. To speculate on the mechanisms underlying the protective role of PPARα agonists in MASLD, address the following recent reference (PMID: 39355870, page 163: " USP29 stabilized ACSL5 expression to promote the activation of fatty acids, which in turn activated PPARa to induce the expression of genes involved in FAO and then directed them to the mitochondria for β-oxidation.").

Author Response

Reviewer 3

We thank Reviewer 3 for the detailed review of our manuscript.

Review comments number 5 and 6 overlapped with the same sentence, so the comment numbers have been reassigned to make nine comments and our responses.

We have addressed your concerns as follows:

The manuscript presents an interesting investigation into pemafibrate's effects on MASLD in both wild-type and eNOS-deficient mice. However, several fundamental issues need to be addressed:

1. In the introduction section, elaborate on the MASLD disease burden by addressing the most updated epidemiology of MASLD (PMID: 39159948). 

(Response) We have cited specified literature and added comments regarding MASLD epidemiology as follows (lines 39–42 in the revised manuscript):

“Metabolic-dysfunction-associated steatotic liver disease (MASLD) currently affects approximately 40% of the global adult population. The prevalence of MASLD among adults is predicted to rise to over 55% by 2040, which will become the most common cause of chronic liver disease worldwide [1-3].”

2. The authors acknowledge that eNOSKO mice did not show the expected worsened MASLD phenotype compared to WT mice. This finding contradicts their initial hypothesis and undermines the study's primary rationale. While they provide potential explanations (common mitochondrial dysfunction mechanisms), this major deviation from expected results requires more thorough investigation and discussion of its implications for the study's conclusions.

(Response) Thank you for pointing out this discrepancy. We were motivated to conduct this study because a previous report showed that the effect of pemafibrate on promoting postischemic angiogenesis was dependent on eNOS activation (Reference No. 21). Therefore, we have revised the Introduction to state that our main purpose in the study was to verify whether the effects of pemafibrate are dependent on the eNOS system, considering the relationship between vascular endothelial function and MASLD, to verify the previously reported effect of pemafibrate on ameliorating MASLD (lines 49-50 and lines 74–81).

“Therefore, pharmaceutical agents that activate eNOS may ameliorate the pathology of MASLD.”

“Furthermore, pemafibrate enhanced ischemia-induced revascularization, which was dependent on eNOS activation [21]. The results together indicate that pemafibrate contributes to inhibiting MASLD progression through activating vascular endothelial function, which is achieved through regulating the eNOS system. Therefore, the purpose in this study was to clarify the molecular mechanism through which pemafibrate ameliorates the pathology of MASLDand to determine whether this effect depends on vascular endothelial function via the eNOS system. “

3. The study uses n=8 per group, but no power calculation or sample size justification is provided. Given the unexpected lack of difference between WT and eNOSKO mice, a post-hoc power analysis would be valuable to determine if the study was adequately powered to detect meaningful differences.

(Response) Thank you for your very thoughtful comment. Calculating statistical power and validating the sample size are important, but we could not perform post hoc tests on the individual outcomes or optimize the sample size for each experiment. The sample size was determined by the study budget. Figure S4 in the revised supplementary materialsshows that if we pre-analyzed the number of samples needed for a one-way ANOVA with four groups using power analysis, we would have needed 45 mice per group, which would have considerably exceeded our available funding and was thus considered unrealistic. We have now described the study limitations in terms of sample size and statistical power in the Experimental Limitations section because these issues must be acknowledged (lines 460–463).

“Second, the results of a power analysis to ensure the detection of differences in the data among the four groups showed that a sample size of 45 mice per group was required (Figure S4); therefore, the statistical power of our study is limited due to the small sample size used.”

4. The pemafibrate dosing (0.17 mg/kg/day) is based on estimated water consumption and body weight averages. This approach introduces variability in actual drug exposure between animals. Individual monitoring of water consumption or alternative delivery methods (e.g., gavage) would provide more precise dosing.

(Response) The material and human resource limitations in our study environment hindered our ability to provide drugs via forced oral administration daily for 10 weeks. We previously conducted an experiment in which we dissolved pitavastatin in a drinking water bottle (PMID: 17967781 and 28592707). The results showed that drug administration was almost fully effective. The drinking water was also individually monitored in our current study, as described in Figure S3 in the revised supplementary materials. We think that the drug exposure did not considerably differ. However, as the reviewer stated, forced oral administration of a fixed drug dose during the experimental period would have ensured consistent drug exposure; therefore, we added an additional description of this aspect as an experimental limitation (lines 453 to 459).

“The following concerns regarding the study methods and analysis should be considered: First, the drinking water was individually monitored, as shown in Figure S3. We do not think that the drug exposure considerably differed among individuals. However, the forced oral administration of a fixed drug dose during the experiment would have ensured consistent drug exposure. Therefore, oral gavage with a fixed pemafibrate dose should be used in a future study to further increase the reliability and reproducibility of our results and minimize interindividual data variability. “

5. The 10-week treatment period begins at 8 weeks of age, ending at 18 weeks. Justification for this specific timing and duration is lacking. Earlier intervention might have revealed different outcomes, particularly given that MASLD pathogenesis begins early in this model.

(Response) Thank you for this feedback. We reviewed previous studies; however, no strong evidence was found showing that a 10-week pemafibrate intervention from 8 to 18 weeks of age would have been appropriate for our study design. In addition, the effectiveness of earlier pemafibrate interventions in ameliorating MASLD must be verified, as noted by the reviewer. Therefore, we have included descriptions of the unresolved issues that should be investigated in future studies (lines 442–445).

“Although we reviewed previous studies, we found no strong evidence that a 10-week pemafibrate intervention from 8 to 18 weeks of age was appropriate for our design. Moreover, whether earlier pemafibrate intervention is more effective in ameliorating MASLD regardless of eNOS gene disruption should be verified.”

6. The selective reduction in Ncf2, Ncf4, Cyba, and Cybb expression without changes in Nox1, Nox4, Rac1, Ncf1, and Noxo1 is intriguing but requires a deeper mechanistic explanation. The authors should discuss why only specific NADPH oxidase subunits were affected and the functional implications of this selective regulation.

(Response) Thank you for your valuable comments that address the core issue in this study. The reason for the difference in the suppressive effect of pemafibrate depending on NADPH oxidase subunit gene expression is unknown. We have revised the text and provided possible explanations for the differences in pemafibrate efficacy (lines 376–381).

“The results showed that the NADPH oxidase system unit genes may be differently activated and inhibited due to differences in metabolic and/or physical stimuli, organ specificity, the presence or absence of PPRE in each subunit promoter region, the recruitment of coactivators involved in activating PPARα, and the effective pemafibrate concentration. Further detailed studies are required to verify this hypothesis.”

7. While pemafibrate increased AMPK phosphorylation regardless of eNOS status, the upstream mechanisms leading to this activation remain unclear. The relationship between PPARα activation and AMPK phosphorylation in this context needs better elucidation.

(Response) The point raised by Reviewer 3 is important in considering the effect of pemafibrate in treating MASLD. However, reports on the relationship between PPARα and AMPK are scarce, and many aspects of this relationship remain unclear. We have now included additional relevant references and have described the possibility that PPARα and AMPK affect each other rather than having unidirectional effects (lines 411 to 420).

“Pemafibrate activates the phosphorylation of AMPK in the renal tissue of diabetic db/db mice [31]. The glucocorticoid receptor and PPARα cooperatively activate a lipid catabolic gene program in primary hepatocytes via directly recruiting of the activated AMPK [45]. The results of the analyses of the protein–protein interaction network, molecular docking, functional enrichment and Western blotting showed that isoxanthohumol, the major hop flavonoid, activated the hepatic phosphorylation of AMPK and PPARα, with the PI3/Akt signaling pathways activated in a hyperlipidemic mouse model [46]. These results suggest that the amelioration of MASLD observed in our study was partly due to reciprocal reactions between phosphorylated AMPKα and activated PPARα via pemafibrate stimulation.”

8. The use of the Kruskal-Wallis test with Dunn's multiple comparisons is appropriate for non-parametric data, but the manuscript lacks information about data distribution testing and assumptions validation.

(Response) Testing the normality of the sampling distribution required a sample size of approximately 30 per group. Therefore, verifying normality with a sample size of eight per group, which we used in this study, was not a critical aspect in this study. Although the statistical power was low, we used the nonparametric Kruskal–Wallis test with Dunn's multiple comparison test to avoid detecting excessive significance and to find only highly reproducible results. Therefore, our response is the same as that to the third point of your concern (lines 460–463).

9. To speculate on the mechanisms underlying the protective role of PPARα agonists in MASLD, address the following recent reference (PMID: 39355870, page 163: " USP29 stabilized ACSL5 expression to promote the activation of fatty acids, which in turn activated PPARa to induce the expression of genes involved in FAO and then directed them to the mitochondria for β-oxidation.").

(Response) Thank you for providing us with useful information . We have cited the reference you suggested in the Introduction section considering the structure of the manuscript, and we incorporated the description of USP29 into the revised manuscript (lines 56 and 61).

“The PPARα plays a role in MASLD pathogenesis due to its role in β-oxidation in the liver. Hu et al. reported that ubiquity-specific protease 29 stabilizes long-chain alkyl-CoA synthesis 5 expression through K48 deuniquitination and promotes fatty acid activation, which in turn activates PPARα and induces the expression of the genes involved in fatty acid β-oxidation [12]. “

Round 2

Reviewer 1 Report

The authors have addressed most of my suggestions. 

It is highly recommended that authors follow the below unattended suggestion, “Subtitles of result section should be written in present tense. This because they are showing the main findings of the investigation, which means that if someone else performs the same experiments by using the same models with the same variables, they will get the same results”.

Author Response

It is highly recommended that authors follow the below unattended suggestion, “Subtitles of result section should be written in present tense. This because they are showing the main findings of the investigation, which means that if someone else performs the same experiments by using the same models with the same variables, they will get the same results”.

(Response) We appreciate your helpful advice. Following Reviewer 1's suggestion, We have changed the subtitle of the results to a present tense in each section.

Reviewer 3 Report

The authors addressed all concerns raised by the reviewer. 

NA

Author Response

We would like to once again thank reviewer 3 for the appropriate advice.